# Complications and Adverse Events of Gonadal Vein Embolization with Coils

**DOI:** 10.3390/jpm12111933

**Published:** 2022-11-20

**Authors:** Sergey G. Gavrilov, Nadezhda Y. Mishakina, Oksana I. Efremova, Konstantin V. Kirsanov

**Affiliations:** Department of Fundamental and Applied Research in Surgery, Pirogov Russian National Research Medical University, 10/5 Leninsky Prospect, Moscow 119049, Russia

**Keywords:** pelvic venous disorder, gonadal vein embolization with coils, complications

## Abstract

**Background:** The efficacy and safety of gonadal vein embolization (GVE) with coils in the treatment of pelvic venous disease (PeVD) has not been fully investigated, and the outcomes after GVE do not always meet expectations of both doctors and patients. The study was aimed at assessing the incidence and causes of the complications after GVE with coils in patients with PeVD. **Methods:** This retrospective cohort study included 150 female patients with PeVD who underwent GVE with coils in 2000–2020. A total of 4975 patients with chronic pelvic pain (CPP) were examined, of which 1107 patients had the PeVD-related CPP and 305 underwent surgical or endovascular interventions on the gonadal veins. Complication rates were evaluated 30 days after GVE and classified according to the Society for Interventional Radiology (SIR) adverse event classification system. The pain severity before and after GVE was assessed using a visual analogue scale (VAS). All patients underwent duplex ultrasound after GVE, while patients with persisting pain syndrome and suspected perforation of the gonadal vein were also evaluated using computed tomographic venography. **Results:** At 30 days after GVE, the CPP was decreased in 109 (72.6%) patients (from 8.2 ± 1.5 at baseline to 1.7 ± 0.8 scores, *p* = 0.0001) and persisted in 41 (27.4%) patients (mean change from 8.1 ± 0.7 at baseline to 7.8 ± 0.4 scores; *p* = 0.71). Post-embolic syndrome (PES) occurred in 22% of patients and was completely resolved in 1 month after GVE. The efficacy of GVE in the CPP relief after resolving PES was 94.6%. The GVE complications were identified in 52 (34.6%) patients. Minor complications included access-site hematoma (4%) and allergic reactions (1.3%), and major complications included protrusion of coils (5.3%), thrombosis of the parametrial/uterine veins (21.3%) and deep veins of the calf (2.7%). **Conclusions:** Gonadal vein embolization with coils in the treatment of PeVD is associated with the development of specific complications and adverse events. The most common complication was pelvic vein thrombosis. Post-embolization syndrome should be considered as an adverse event of this procedure.

## 1. Introduction

Gonadal vein embolization (GVE) with nitinol or platinum coils is widely used in the treatment of PeVD caused by the valvular incompetence of gonadal, parametrial, and uterine veins [1,2]. Most authors report a high efficacy of this technique in reducing blood flow through the gonadal veins (GV) and relieving symptoms of the disease [3,4,5]. In the Society for Vascular Surgery (SVS) and American Venous Forum (AVF) guidelines, GVE is considered the standard of treatment for PeVD with a grade of recommendation 2B, due to the moderate quality of evidence [6]. Moreover, other studies report wide variability in the GVE outcomes in terms of pelvic venous pain (PVP) elimination, persistence or intensification of pain after GVE, and coil migrations and protrusions [7,8,9,10]. It is known that 6% to 32% of patients do not achieve significant pain relief after the procedure [11]. Most studies of GVE in the treatment of PeVD are characterized by only a statement of the fact of any complication without investigating the causes of its development [3,4,8,9]. At the same time, it is well known that it is a thorough study of complications that makes it possible to avoid failures in the future, to improve the therapeutic technique, or to abandon its use altogether.

The aim of this study was to assess the incidence of GVE complications in patients with PeVD and to study the causes of their development based on 20 years of experience in using this technique in clinical practice.

## 2. Methods

This retrospective cohort study included 150 female patients with PeVD who were treated at the Savelyev University Surgical Clinic of the Pirogov Russian National Research Medical University in the period from 2000 to 2020. The study was approved by the local ethics committee of the University and registered at ClinicalTrials.gov (ID: NCT05085938). Patient informed consent was not required due to the retrospective nature of this study.

The inclusion criteria were: (1) the presence of symptoms and signs of PeVD (pelvic venous pain, dyspareunia, discomfort/heaviness in the hypogastric region, vulvar varicosities); (2) reflux in the gonadal (GVs), parametrial (PVs) and/or uterine veins (UVs), according to duplex ultrasound (DUS) and computed tomographic (CT) venography; and (3) an isolated coil embolization of GVs.

The exclusion criteria were as follows: the presence of nutcracker and May-Thurner syndromes confirmed by ultrasound and contrast X-ray imaging techniques; open, endoscopic, or hybrid interventions on the gonadal and iliac veins.

The diagnosis of PeVD was verified using transvaginal and transabdominal DUS, multiplanar computed tomographic venography (MSCT), ovarian and pelvic venography (OPV). A detailed description of these techniques is provided elsewhere [12,13,14].

### 2.1. Patients

A total of 4975 female patients with CPP aged from 18 to 69 years (interquartile range 17 years) were examined in the period from 2000 to 2020, of whom 1620 (32.5%) women had pelvic varicose veins with pathological reflux, according to DUS. The pelvic vein dilation and reflux were the only cause of PVP in 1107 (22.2%) patients. Competing gynaecological, urological or neurological diseases were diagnosed in 513 (10.3%) patients.

Patients with an obvious venous cause of CPP (i.e., pelvic pain associated with varicose veins and reflux in them, according to transvaginal and transabdominal DUS, no competing pathology) were treated in the surgical clinic. Patients with isolated dilatation and reflux in the parametrial and uterine veins (***n*** = 802) received only medical treatment with venoactive drugs (VADs). Surgical or endovascular treatment was performed in 305 patients and included open extraperitoneal GV resection in 92, endoscopic GV resection in 30, and GVE in 160 patients. In 23 (2.3%) patients, a combination of May-Turner syndrome and pelvic congestion syndrome (PCS) was diagnosed. Of them, 8 patients underwent only iliac vein stenting, 10 were treated with GVE in combination with iliac vein stenting, and 5 refused endovascular intervention. According to the inclusion/exclusion criteria, 150 patients who underwent GVE with coils and had the necessary information in their medical records were eligible for the analysis. The severity of PVP, dyspareunia and pain before and after GVE was assessed using a visual analogue scale (VAS) ranging from 0 (no pain) to 10 (worst possible pain). The flow chart of the study is presented in Figure 1.

### 2.2. Gonadal Vein Embolization with Coils

Indications for GVE with coils were the presence of PeVD symptoms (CPP, dyspareunia, discomfort or heaviness in the hypogastric area), reflux (>1 s) in GV, PV, UV single-trunk conductive type of GV, the GV diameter < 10 mm, according to transvaginal and transabdominal DUS, and the absence of nutcracker or May-Thurner syndromes, according to DUS, renal venography or multiplanar pelvic venography.

GVE was performed under local anaesthesia with 5.0–10.0 mL of 0.5% lidocaine solution with a patient under intravenous sedation. For the left GV embolization, the transfemoral approach (119 patients) was used, while for the right or both GV embolization, the transjugular approach (31 patients) was used. The vein puncture was performed under ultrasound guidance. The 5F multipurpose angiographic catheters (Radiofocus, Terumo Europe, Leuven; Belgium), standard ‘moving core’ J 0.035” guidewire, and an angled hydrophilic guidewire (Radiofocus; Terumo Corp., Japan) were used. For the GV occlusion, the pushable 0.035” standard stainless-steel coils (Gianturco; William Cook, Bjæverskov, Denmark) and 0.035” coils made of Inconel with interwoven long collagen fibrils (MReye; Cook Medical Inc., Bloomington, IN, USA) were used. The diameter of coils was 8–12 mm, and the length was 10–20 cm. In this study, GVE was not combined with sclerotherapy of GVs.

### 2.3. Assessment of GVE Complications

Complications of GVE were graded using the Society for Interventional Radiology (SIR) adverse event classification system [15]: A—No therapy, no consequences; B—Nominal therapy, no consequence; includes overnight admission for observation only; C—Requires therapy, minor hospitalization (<48 h); D—Requires major therapy, unplanned increase in level of care, prolonged hospitalization (>48 h); E—Permanent adverse sequelae; F—Death. Minor complications include classes A and B, and major complications include classes C to F.

In a standard case, DUS of the veins of pelvis and lower limbs was performed in all patients on the next day and day 30 after the procedure. In case of persistent pelvic pain or the occurrence of hyperthermia and pain in the area of the embolized vein, DUS was performed on days 1, 3, and 10 after GVE. Patients with persistent pain syndrome underwent additional MSCT, OPV within 10 days.

### 2.4. Statistical Methods

Statistical analysis was performed using Microsoft Excel (Microsoft Corp, Redmond, WA, USA), Statistica 10 (StatSoft, TIBCO, Palo Alto, CA, USA) and VassarStats online calculator (open source online project). Results are presented as quantitative and categorical variables. For quantitative variables, the Mann-Whitney test was used; for categorical variables, the Chi-square test was used. The mean, standard deviation (M ± SD), odds ratio with 95% confidence interval, Students’ *t*-tests were calculated. Differences were considered statistically significant at a *p* value less than 0.05.

## 3. Results

Left-sided embolization of GV was performed in 119, right-sided in 7, and bilateral in 24 patients. Duration of GVE ranged from 20 to 55 min and was, on average, 22.5 ± 1.3 min for unilateral and 41.3 ± 4.2 min for bilateral embolization, accordingly. For occlusion of one gonadal vein, 3 to 8 coils were used (mean 5.2 ± 1.4 coils). The mean volume of contrast media was 30.2 ± 3.2 mL for unilateral and 48.5 ± 5.1 mL for bilateral GVE.

Technical success (elimination of the blood flow in the gonadal veins) was achieved in 100% of patients. At 30 days after GVE, the pelvic pain had decreased or completely relieved in 109 (72.6%) patients (from 8.2 ± 1.5 at baseline to 1.7 ± 0.8 scores, *p* = 0.0001) and persisted at the same level or increased in 41 (27.4%) patients (mean change from 8.1 ± 0.7 at baseline to 7.8 ± 0.4 scores; *p* = 0.71).

Of 150 patients, the GVE complications were reported in 52 (34.6%) patients (8 had minor and 44 had major complications). Minor complications (classes A and B) included access-site hematoma and allergic reactions to the contrast agent, and major complications (classes C and D) included thrombosis of the parametrial, uterine veins and deep veins of the calf or protrusion of coils. Postembolization syndrome (PES) was identified in 22% of the patients and was considered as an adverse event after GVE with coils (Table 1).

Demographic data and baseline clinical characteristics of patients with and without GVE complications are presented in Table 2.

There were no significant differences between the groups at baseline. The analysis of GVE complications revealed some important patterns.

### 3.1. Access-Site Hematoma

This rare complication was observed in 4% of patients. In 3 cases, hematoma occurred after right-sided transfemoral access, and in 1 case after transjugular access. This complication did not require any additional treatment.

### 3.2. Allergic Reactions

Dyspnea, palpitation, arterial hypotension and nausea were reported in 2 (1.3%) patients after left-sided GVE occlusion and control ovarian venography, which was regarded as an allergic reaction to a contrast agent. In one patient, five Gianturco coils were implanted and Omnipaque media was used for venography; in another patient, four MReye coils were implanted and Ultravist media was used for venography. In both patients, the volume of contract agent did not exceed 40 mL. These symptoms were stopped immediately by intravenous administration of glucocorticoids (prednisolone, dexamethasone). The patients had no known history of hypersensitivity to an iodine-containing agent or nickel alloy. The medical records did not contain information about further examination of these patients for hypersensitivity to iodine and metals. In the postprocedural period, no symptoms of allergy were reported, and PES was absent in both patients.

### 3.3. Thrombosis of the Parametrial Veins (PV), Uterine Veins (UV) and Deep Veins of the Calf (DVC)

On the next day after GVE, thrombotic lesions of the pelvic veins were found by DUS in 32 (21.3%) patients: 11 had thrombosis of the left PV after left-sided GVE, 19 had bilateral PV thrombosis after bilateral GVE, and 2 had thrombosis of PV and UV after bilateral GVE. Therefore, the procedure was complicated by PV and/or UV thrombosis in 21 (87.5%) out of 24 patients who underwent bilateral embolization.

Thrombosis of DVC was revealed in 4 (2.7%) patients on the next day after the left-sided GVE. A characteristic feature of postembolization thrombosis of PV and UV was the absence of any specific clinical manifestations of this complication; patients did not notice an increase in pelvic pain or severe fever. On the contrary, patients with DVC thrombosis complained of pain in the lower leg, ankle edema, and had Homans and Moses signs in the physical examination. All patients with thrombosis of PVs, UVs and calf deep veins received anticoagulant therapy with unfractionated heparin (UFH) 450 U/kg TID or low molecular weight (LMWH) 1 mg/kg BID subcutaneously into the abdominal wall for 1–2 weeks, in the settings of vascular department, with the following administration of indirect anticoagulants (warfarin, rivaroxaban) for 3 months. In the acute phase of thrombosis, none of the patients showed an increase in the grade of thrombosis, involvement of other veins of the pelvis or lower extremities in the thrombotic process, or the development of pulmonary embolism.

### 3.4. Postembolization Syndrome (PES)

This condition was recorded in 33 (22%) patients after left-sided (28 patients) or bilateral (5 patients) GVE. PES was characterized by the occurrence of pain along the embolized veins in isolation (21 patients) or in combination with increased pelvic pain (12 patients), fever up to 37.4–37.9 °C, fatigue, and malaise (Table 3).

The development of PES was observed with the use of both Gianturco coils (12.7%) and MReye coils with interwoven long collagen fibrils (9.3%, *p* = 0.46). The PES duration ranged from 5 to 23 days (mean 17.3 ± 3.2 days). Computed tomography performed in 25 patients with PES 5 to 7 days after GVE indicated the absence of perforation of the gonadal veins or retroperitoneal hematoma. The treatment of patients with PES was carried out in the vascular department. For relieving the symptoms, nonsteroidal anti-inflammatory drugs (diclofenac 150 mg/day) and venoactive drugs (micronized purified flavonoid fraction, 1000 mg/day) were used. Seven patients with PV thrombosis, which was diagnosed using DUS on the next day after GVE, received anticoagulant therapy with UFH or LMWH. In all patients, medical treatment was able to significantly reduce or completely relieve the PES symptoms. The GVE efficacy in the CPP relief after resolving PES was 94.6%. Thus, the development of PES should be considered as an adverse event after GVE with coils.

### 3.5. Coil Protrusions

This complication occurred in 8 (5.3%) patients, including one patient with PV thrombosis revealed by DUS. All patients with coil protrusions underwent left-sided GVE. A characteristic feature of these patients was a low body mass index (BMI) (mean 18.3 ± 0.5 kg/m^2^). They all had intense (up to 7–9 VAS scores) pain along the embolized vein for more than 30 days after the procedure. The use of anti-inflammatory and venoactive drugs had no significant success, and the pain syndrome was resistant to the therapy. The follow-up DUS, MSCT, ovarian and pelvic venography indicated the absence of perforation of the gonadal veins or any abdominal or vascular accident that could explain persistent pain. These factors substantiated the need for performing open retroperitoneal (in 7 patients) or endoscopic retroperitoneal (in 1 patient) excision of the left gonadal vein with coils within 32 to 45 days (mean 34.3 ± 3.8 days). During surgery, the protrusions of the whorls of coils, along with a low amount of retroperitoneal fatty tissue, were observed (Figure 2*)*.

After surgery, all patients noted a significant decrease in pain syndrome within 3–5 days and its complete relief by day 30 after surgery.

## 4. Discussion

The efforts to minimize surgical trauma and improve the aesthetics of PeVD treatment led to the development and widespread use of GVE with coils [2,3,4,16,17,18,19]. The overwhelming majority of the authors of these studies indicate a 100% elimination of blood flow in the gonadal veins and a very good aesthetic result. Indeed, analysis of literature data indicates that GVE relieves PeVD symptoms in about 70% of patients [11]. At the same time, there is an increase in the number of publications drawing attention to severe complications of GVE that require serious medical, endovascular or surgical treatment [9,20,21,22,23,24,25]. In most cases, these studies are presented as clinical case reports, which creates the illusion of a rare occurrence of complications of GVE with coils. The present study refutes this misconception.

In this retrospective study, GVE complications were found in 56.6% of patients; the proportion of severe complications in the studied cohort was 51.3%, i.e., in more than half of the patients. A legitimate question arises as to whether GVE with coils is safe and feasible for widespread use in the treatment of PeVD? The answer should be sought in the reasons for the development of GVE complications.

Potential causes for access-site hematoma may be the use of puncture and catheterization of large venous trunks (femoral and jugular veins), adequate compression of which with a bandage (especially the jugular vein) is not always possible. One of the options for preventing the access-site hematoma is the use of cubital approach and minimization of trauma to the venous wall by using 0.018′’ microcatheters and platinum microcoils.

A rare complication of GVE was allergic reaction to the contrast agent (in 1.3% of patients). Preoperative testing of patients for hypersensitivity to iodine-containing substances and nickel alloys is probably the only measure for preventing allergic reactions during and after GVE [26,27], although such a recommendation is not included in the guidelines [5,6,28]. The large cohort studies are warranted to determine the need for such a routine testing prior to venography and GVE.

The development of PV, UV and DVC thrombosis in 24% of patients after GVE can be explained from several points of view. First, a sudden cessation of blood outflow through one or two venous collectors on the top of varicose transformation of PV and UV after GVE is accompanied by an aggravation of venous stasis in the visceral veins of the pelvis. Deposition of blood in the PV and UV and their varicose transformation can be considered the leading factors of postprocedural thrombosis. Despite the absence of any research on this issue, it is likely that the use of prophylactic doses of anticoagulant drugs will avoid or reduce venous thromboembolism (VTE) after GVE. In addition, the implantation of a foreign metal agent into the lumen of the vessel also serves as a factor that activates a cascade of pathological changes in the blood coagulation system, which is another reason for the use of anticoagulant prophylaxis of VTE after GVE.

Mechanical contact of the coil with the wall of the gonadal vein is inevitably accompanied by an inflammatory reaction [29]. In addition, the diameters of coils used for GVE exceed the diameter of the vessel by 20%, which can cause the whorls of coil to contact the surrounding tissues, in particular, the genitofemoral nerve. PES occurred in 22% and was absent in 78% of patients. The reason for this is not entirely clear, because the technique for performing the procedure was the same in all cases, and the type of coils did not influence the incidence of PES. A significant difference in the BMI between patients with and without PES (20.8 ± 0.9 kg/m^2^ vs 25.3 ± 1.4 kg/m^2^, *p* = 0.007) suggests that the amount/thickness of retroperitoneal adipose tissue surrounding the gonadal vein may be critically small to serve as a damper barrier between the embolized vein and the genitofemoral nerve [9]. Therefore, it is logical to assume that these are factors that cause postembolization pain in every fifth patient undergoing GVE. At the same time, the GVE efficacy in the CPP relief after resolving PES was 94.6%. Thus, it can be argued that PES is a recoverable condition, which aggravates the course of the post-embolization period and can result in the persistence of the pain syndrome. PES can be considered as an adverse event potentially curable within 1 month after GVE, that has no significant effect on the long-term treatment outcomes.

The same mechanism of the development of intractable pain in the area of the embolized vein can be considered in case of coil protrusions in 5.3% of patients with a low BMI (18.3 ± 0.5 kg/m^2^), as evidenced by the intraoperative findings during endoscopic resection of the left gonadal vein with coils. The odds ratio of developing PES and coil protrusions in patients with a BMI less than 19 kg/m^2^ is 12 times higher than in patients with a BMI greater than this value (OR 12.4; 95% CI: 3.6–42.7).

Therefore, GVE with coils in lean patients cannot be considered the optimal treatment. Probably, the use of endovenous chemical ablation of the gonadal veins (such as cyanoacrylate glue) will significantly reduce the number of postembolization complications [30,31].

### Limitations

The present study has several limitations. First, this is a retrospective study evaluating the results of previous endovascular interventions on the gonadal veins, and although the criteria for determining indications for GVE and the choice of coils were, in general, standardized, one cannot rule out the preferences of the operating surgeon that could affect the outcome of embolization. Second, a significant cohort of patients was excluded from the study due to insufficient data in their medical records, the presence of concomitant pathology of the veins and pelvic organs, and the performance of simultaneous endovascular procedures on the gonadal and iliac veins. In addition, no baseline or history data on the presence of hypersensitivity to metals and iodine-containing drugs were available, which excludes this factor from the analysis in the study.

Despite these limitations, the study results can be used to define indications for the endovascular treatment of PeVD, the differentiated choice of embolization agents and systems for their delivery to the gonadal veins, and the justified rejection of the use of GVE with coils in a patient with PeVD.

## 5. Conclusions

Despite the successful use of GVE with coils in the treatment of PeVD, the post-embolization period is accompanied by the frequent development of complications and adverse events. Significant, severe complications of GVE with coils are pelvic vein thrombosis and coil protrusions. Post-embolization syndrome should be considered as an adverse event, which does not affect the GVE efficacy. Further prospective comparative studies are warranted for the development of a differentiated approach to the choice of procedure to reduce blood flow through the gonadal veins and for evaluation of the efficacy and safety of new embolic agents in the treatment of patients with PeVD.

## Figures and Tables

**Figure 1 jpm-12-01933-f001:**
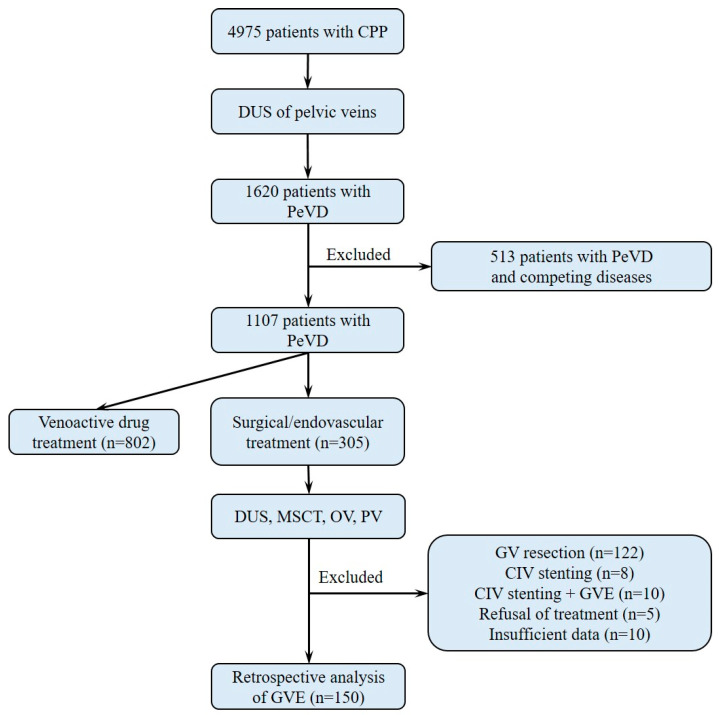
The study design flowchart. Abbreviations: CIV, common iliac vein; CPP, chronic pelvic pain; GV, gonadal veins; GVE, gonadal veins embolization; GVR, gonadal veins resection; DUS, duplex ultrasound; MSCV, multislice computed venography; OV, ovarian venography; PeVD, pelvic venous disorder; PV, pelvic venography.

**Figure 2 jpm-12-01933-f002:**
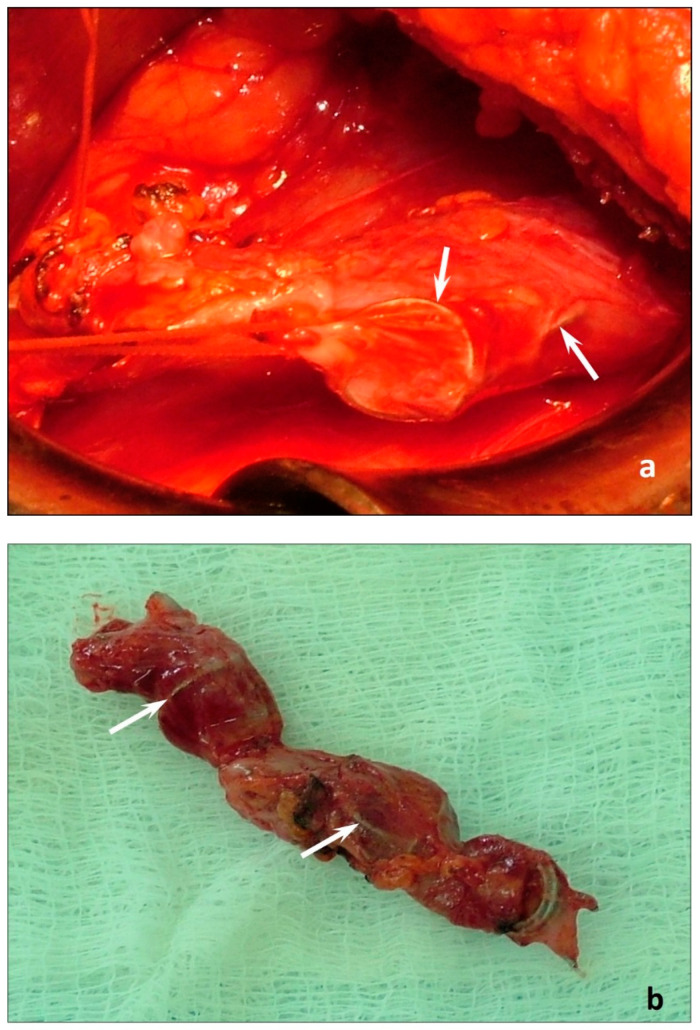
Photos of the stage of extraperitoneal excision of the left gonadal vein with coils (**a**) and the removed vein with coils (**b**). The coil protrusions are indicated by arrows.

**Table 1 jpm-12-01933-t001:** Complications and adverse events of GVE with coils in patients with PeVD (*n* = 150).

Complication/Adverse Event	Events, n (%)
Access-site hematoma	6 (4)
Allergic reactions	2 (1.3)
Thrombosis of pelvic veins (PV, UV)	32 (21.3)
DVC	4 (2.7)
PES	33 (22)
Coil protrusion	8 (5.3)

Abbreviations: DVC, deep veins of the calf; GVE, gonadal vein embolization; PeVD, pelvic venous disease; PES, postembolization syndrome; PV, parametrial veins; UV, uterine veins.

**Table 2 jpm-12-01933-t002:** Demographic data and baseline clinical characteristics of patients who underwent GVE (*n* = 150).

Parameter	GVE Complications		*p* Value
	Yes, *n* = 65	No, *n* = 85	
Age, M ± SD, years	29.3 ± 1.7	28.5 ± 2.2	0.77
BMI, M ± SD, kg/m^2^	24.5 ± 2.1	22.4 ± 1.1	0.37
Number of pregnancies, n	1–7	1–6	-
Number of births, n	1–4	1–5	-
Known allergy to metals and contrast agents, n (%)	0	0	-
Disease duration, M ± SD, years	5.7 ± 2.1	5.5 ± 1.3	0.56
PVP, n (%)	65 (100)	85 (100)	-
Severity of PVP before GVE, VAS scores	8.2 ± 1.5	8.1 ± 0.7	0.45
Dyspareunia, n (%)	55 (84.6)	69 (81.2)	0.69
Heaviness in the hypogastrium, n (%)	65 (100)	85 (100)	-
Vulvar varicosities, n (%)	12 (18.4)	14 (16.5)	0.47
Concomitant diseases			
Lumbosacral osteochondrosis, n (%)	2 (3)	1 (1,2)	0.13
Chronic gastritis, n (%)	4 (6.2)	5 (5.8)	0.82
Cholelithiasis, n (%)	3 (4.6)	1 (1.2)	0.11
Small size uterine fibroids	3 (4.6)	2 (2.4)	0.37
VVLE, n (%)	13 (20)	11 (12.9)	0.44
CVD of CEAP class C2-C3, n (%)	19 (29.2)	22 (25.9)	0.12

Abbreviations: BMI, body mass index; CEAP, Clinical, Etiologic, Anatomic, and Pathophysiologic classification; CVD, chronic venous disease; GVE, gonadal vein embolization; PVP, pelvic venous pain; VAS, visual analogue scale; VVLE, varicose veins of the lower extremities.

**Table 3 jpm-12-01933-t003:** Clinical and morphological parameters of patients at 3 days after GVE and characteristics of endovascular intervention (*n* = 150).

Parameter	Patients without PES (*n* = 117)	Patients with PES (*n* = 33)	*p* Value
Age, M ± SD, years	28.7 ± 1.4	29.3 ± 1.1	0.37
BMI, M ± SD, kg/m^2^	25.3 ± 1.4	20.8 ± 0.9	0.007
Disease duration, M ± SD, years	5.5 ± 1.8	5.3 ± 1.5	0.93
PVP before GVE, M ± SD, VAS scores	8.3 ± 0.5	8.1 ± 0.7	0.81
PVP after GVE, M ± SD, VAS scores	4.7 ± 0.3	7.8 ± 0.4	0.0001
Pain along the embolized vein, n (%)	0	33 (100)	-
Increase in pelvic pain, n (%)	0	12 (36.4)	-
Fever, n (%)	0	33 (100)	-
Fatigue, malaise, n (%)	0	33 (100)	-
PV and/or UV thrombosis, n (%)	25 (21.3)	7 (21.2)	0.98
Calf DVT	4 (3.4)	0	-
Diameters of the gonadal veins			
Left gonadal vein, mm	7.9 ± 0.8	8.1 ± 0.6	0.84
Right gonadal vein, mm	7.5 ± 0.2	7.2 ± 0.3	0.40
Side of embolization			
Left-sided, n	91 (77.8)	28 (84.8)	0.09
Right-sided, n	7 (5.9)	0	-
Bilateral, n	19 (16.3)	5 (15.2)	0.11
Number of coils			
Left-sided, n	5.7 ± 1.2	6.4 ± 1.6	0.72
Right-sided, n	5.2 ± 0.5	0	-
Bilateral, n	9.1 ± 1.7	10.2 ± 1.1	0.58
Type and form of coils			
Gianturco, pushable, helical shapes, n *	35 (23.3)	19 (12.7)	0.11
Size of Gianturco coils, M ± SD, mm	10.4 ± 0.8	11.3 ± 0.5	0.34
MReye, pushable, helical shapes, n *	82 (54.7)	14 (9.3)	0.002
Size of MReye coils, M ± SD, mm	11.2 ± 0.3	11.7 ± 0.4	0.31

Abbreviations: BMI, body mass index; CVD, chronic venous disease; DVC, deep veins of the calf; GVE, gonadal vein embolization; PV, parametrial veins; PVP, pelvic venous pain; UV, uterine veins; VAS, visual analogue scale;.* Number of patients with implanted coils of this type.

## Data Availability

Data available from the corresponding author upon reasonable request.

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
