# Peer review of "Complications and Adverse Events of Gonadal Vein Embolization with Coils"

_jpm, 2022, doi:10.3390/jpm12111933_

Round 1

Reviewer 1 Report

1886551 - Complications of gonadal vein embolization with coils: causes and consequences

Thank you for asking me to review this paper. For the purposes of transparency, I have previously reviewed this paper for another journal.

Unfortunately, rather than address my previous criticisms, the authors have submitted the same paper to a different journal without any major modifications.  

The paper has been written by a doctor with an excellent reputation for publishing in this area. However, this current paper has been written in a highly opinionated and biased manner against the use of coil embolization of the gonadal veins.

Although that might be acceptable in an opinion article, this current work has been presented as a research paper. The authors are reporting their own results, and their experience is certainly not that found commonly amongst those of us who perform this procedure regularly on large numbers of patients with good follow-up.

The authors are clearly getting bad results and feel that this is not a good technique in their own hands. However, rather than saying that, they try to advance the opinion the whole procedure is at fault, rather than considering that their results might be due to their own technique.

The fact that others (such as our own unit) have 2 decades of experience in this area and do not see the high number of severe complications that are reported by the authors, supports this point. Even the authors themselves quote in the discussion that  “The overwhelming majority of the authors of these studies indicate a 100% elimination of blood flow in the gonadal veins and a very good aesthetic result. Indeed, analysis of literature data indicates that GVE relieves PeVD symptoms in about 70% of patients”. This in itself shows that the authors are not getting as good results as others working in the field.

They continue “In most cases, these studies are presented as clinical case reports, which creates the illusion of a rare occurrence of complications of GVE with coils. The present study refutes this misconception.” This is incorrect as the case reports that they quote are indeed case reports, because such coil embolization (a real complication) is rare. It almost always happens when doctors start doing Pelvic Vein Embolization and use non-retractable smooth coils. Most practitioners soon learn to use retractable fibred coils, which virtually eliminates this complication. Moreover, cases where coils that have embolised are removed by open chest or heart surgery suggests that they are being done in units without the expertise to recover such coils with endovenous snares.  

Hence their generalized comments about this technique are not acceptable in the face of world literature from those practitioners who are getting good results with very low complication rates and high proportions of happy patients.

Furthermore, as pointed out below, the two major “complications” noted (post-embolization syndrome and thrombosis of the uterine veins) are not complications but are expected processes after an embolization procedure. For the authors to make these out to be significant complications that should make the world reconsider performing this procedure, they would need to show an effective alternative.   

I have only made comments on the abstract at present for 2 reasons - firstly, they are not supportable at the current time and are highly biased and secondly, the abstract is very important as this is what most people read - and so must give an accurate impression of the work. 

There are several other points in the paper that would need to be addressed if the editors wish to publish the paper.

Abstract:

Introduction

" The efficacy and safety of gonadal vein embolization (GVE) with coils in the treatment of pelvic venous disease (PeVD) has not been definitively determined, and the outcomes of using this technique do not meet the requirements of doctors and patients" extremely biased comment and not substantiated in the literature. There are a great many papers demonstrating the efficacy of coil embolization, the patient improvement in symptoms and also the safety of the technique, even when performed under local anaesthetic and also in the long term.

Results.

The "complications" are worthy of note:

"protrusion of coils (5.3%)" - this is an amazingly high statistic - and something we have never seen in thousands of GVEs - this must be related to the technique used in the author's unit or the device they used and should not be generalized as a widespread complication.

"postembolization syndrome (22%)" is a transient problem that is part of the healing process and does not last into the long term. It is an effect of the inflammation caused by thrombosing veins with coils and is not a complication but a consequence of treatment. Our patients have this included in their consent form and so are expecting it. The discomfort of healing is natural and expected and found in most (if not all) effective procedures - it is not a complication.

"thrombosis of the parametrial, uterine veins" - this is the purpose of GVE. It is also why so many workers in the field use foam sclerotherapy in these vein before deploying coils in the larger veins.

Conclusion:

"GVE with coils in the treatment of PeVD is associated with a high incidence of complications" - this is not the experience of many of us with many years of experience and high numbers of treatments. If the authors have found this in their own practice, they need to qualify their statement: ie: "In our practice, GVE with coils in the treatment ....."

"which can be reduced" is an unproven statement unless supported by randomised controlled trials. The authors will need to produce future work to show that their new approaches can indeed improve on coil embolization.

Author Response

Reviewer #1

The authors are thankful to the Reviewer for a thorough review of the manuscript. Please find our answers below.

Thank you for asking me to review this paper. For the purposes of transparency, I have previously reviewed this paper for another journal.

Unfortunately, rather than address my previous criticisms, the authors have submitted the same paper to a different journal without any major modifications. 

Response: The authors remember very well your review of the abstract our study, and it was exactly the same as this time. Unfortunately, the editors of the previous journal did not provide us with an opportunity to respond to your comments, and we considered it incorrect to hold discussions outside the scope of the reviewing process. But now, thanks to the editors of the Journal of Personalized Medicine, the authors can fully respond to your comments.

The paper has been written by a doctor with an excellent reputation for publishing in this area. However, this current paper has been written in a highly opinionated and biased manner against the use of coil embolization of the gonadal veins.

Response: In fact, this is a team scientific work and all authors contributed to it, although the idea of ​​the study belongs to Professor Gavrilov.

Let the authors disagree with your statement. This manuscript presents only the results of a 20-year retrospective evaluation of gonadal vein embolization (GVE) with coils, and it is based only the statement of facts that are devoid of emotions, so there is no overconfidence in it. Without a doubt, this manuscript is not intended to discredit coil embolization. On the contrary, we considered it necessary to demonstrate that in some patients GVE is an effective treatment, but in another, smaller proportion of patients, GVE should be either not used or used with different embolic agents (Glubran 2, for example). The gonadal vein resection can be used, for example, in patients with a low body mass index. We made such assumptions (please note, only assumptions) that can be confirmed or refuted by further research. Such studies are already ongoing.

Although that might be acceptable in an opinion article, this current work has been presented as a research paper. The authors are reporting their own results, and their experience is certainly not that found commonly amongst those of us who perform this procedure regularly on large numbers of patients with good follow-up.

Response: Of course, this is a research work carried out by a team of authors who have the same point of view on the problem. The authors do not think their findings differ significantly from the global estimates of the efficacy of GVE with coils in relieving pain. According to our study, the clinical efficacy of GVE (CPP relief) is 72.6% 1 month after the intervention. In this study, the outcomes and complications of the procedures were evaluated at 1 month. The authors did not plan to report further observations in this particular study. According to other studies, the GVE efficacy in eliminating CPP varied from 37 to 100%, and the average figure was 70%. This point is discussed in the Introduction and Discussion sections. Our result is 73%, i.е. in line with global estimates.

The authors are clearly getting bad results and feel that this is not a good technique in their own hands. However, rather than saying that, they try to advance the opinion the whole procedure is at fault, rather than considering that their results might be due to their own technique.

Response: The study showed good immediate results. Thus, the clinical efficacy of GVE in the CPP relief was 72.6% at 1 month after the intervention. One month is the period during which the results and complications of the procedure were evaluated in this study. In this study, the outcomes and complications of the procedures were evaluated at 1 month.

The authors believe that the reviewer's statement that we can't perform GVE well is not acceptable. The technical success of GVE according to our data was 100%. There was not a single case of coil migration, venous wall perforation or lesion of non-targeted vessels. The procedure was performed by highly qualified specialists with 10 or more years of experience in the interventional surgery. It is not the procedure and not the doctor's hands that are to blame, but the criteria for its use, which need to be reviewed.

Embolization is a routine procedure in our clinic, and it is performed not only for the gonadal vein occlusion, but in many other urgent and routine settings, and the intervention technique does not present any difficulties for our specialists.

The authors once again point out that the results of the study are in line with global findings: the CPP relief in the early post-procedural period was achieved in 72.6% patients, and the average global rate is about 70%.

The fact that others (such as our own unit) have 2 decades of experience in this area and do not see the high number of severe complications that are reported by the authors, supports this point. Even the authors themselves quote in the discussion that  “The overwhelming majority of the authors of these studies indicate a 100% elimination of blood flow in the gonadal veins and a very good aesthetic result. Indeed, analysis of literature data indicates that GVE relieves PeVD symptoms in about 70% of patients”. This in itself shows that the authors are not getting as good results as others working in the field.

Response: Would you be so kind to provide links to the experience of your clinic, to make it possible to compare our data with yours? Our data are consistent with findings from the world literature, i.e. we achieved technical success in 100% of cases and CPP relief in the immediate post-procedural period in 73% patients. Moreover, this rate increased after the PES manifestations were resolved, and this is obvious. But this happens after resolving PES. The manuscript clarified that after resolving PES, the GVE efficacy increased to 94.6%. However, this does not ignore the fact PES occurred in 22% patients after GVE.

They continue “In most cases, these studies are presented as clinical case reports, which creates the illusion of a rare occurrence of complications of GVE with coils. The present study refutes this misconception.” This is incorrect as the case reports that they quote are indeed case reports, because such coil embolization (a real complication) is rare. It almost always happens when doctors start doing Pelvic Vein Embolization and use non-retractable smooth coils. Most practitioners soon learn to use retractable fibred coils, which virtually eliminates this complication. Moreover, cases where coils that have embolised are removed by open chest or heart surgery suggests that they are being done in units without the expertise to recover such coils with endovenous snares. 

Response: Following your logic, the real complications are only coil migration and venous wall perforation. And the coil migration is, in fact, means nothing. However, we had no such complications, fortunately, unlike other authors.

We have been using fibred coils for the past 12 years, and in the Metods section and in Table 3 it is indicated that we used MReye coils. The same table shows that the type of coils had no effect on the development of PES.

If we follow the definition of complications, this is any deviation from the normal course of the postoperative / post-procedural period. And if 78% of patients had no PES, then 22% had it. This means that GVE is not obligatory associated with PES, and this is obvious.

If we take the case reports on coil migration and protrusion, one can note a small number of patients who underwent GVE. Few patients means few complications. We summarized our 20-year experience of investigating complications in patients after GVE with coils, except for those who underwent simultaneous interventions.

You point out that the coil removal by surgery is due to the inability of the authors of these reports to use endovenous snares to remove them. Does this change the nature of coil migration? It's still a serious complication. However, there was no such complication in our study.

Hence their generalized comments about this technique are not acceptable in the face of world literature from those practitioners who are getting good results with very low complication rates and high proportions of happy patients.

Response: There is no evidence for such a statement. The authors once again focus the reviewer's attention on the fact that the GVE efficacy was 73% after 1 month. Both smooth coils and fibred coils were used in the work, and this did not influence the incidence of PES (see Table 3). No matter what type of coil is used, as the venous wall response to it will be the same. This is a pathophysiological response, and it can be strong (with the development of PES) or weak (without PES).

As for the world experience, we report that the study of Monedero et al https://doi.org/10.1258/026835506775971108  showed that PES after GVE was detected in 12.5-54% (depending on clinical manifestations). The study of De Gregorio et al https://doi.org/10.1016/j.jvir.2020.06.017 PES revealed PES in 10.3% of patients, but the authors also observed coil migrations (in 1.9% cases) and access-site hematomas (in 3.7% cases). And these authors regarded PES as a complication.

Our experience and literature data do not contradict each other, and our data are quite reasonable.

The reviewer tries to present the authors of the manuscript as doctors who do not know how to perform this procedure, but write about it. This is unacceptable and does not meet any ethical standards.

Our clinic has extensive experience in treating patients with PeVD using endovascular and surgical methods. This is confirmed by publications in reputable scientific journals: Journal of vascular surgery, venous and lymphatic disorders, Journal of Personalized Medicine, Phlebology, and Diagnostics.

Furthermore, as pointed out below, the two major “complications” noted (post-embolization syndrome and thrombosis of the uterine veins) are not complications but are expected processes after an embolization procedure. For the authors to make these out to be significant complications that should make the world reconsider performing this procedure, they would need to show an effective alternative.  

Response: The "expected process" is a highly dubious term. Wound infectious complications, thrombosis at the anastomosis site or shunt thrombosis after arterial reconstruction, hematoma at the site of vessel puncture, thermally-induced thrombosis of the femoral vein after radiofrequency ablation of the great saphenous vein, etc. are also expected processes, but they are also considered as complications of interventions, if they occur. Why did GVE with coils become an exception?

The authors offer the reviewer a consensus solution for this controversial issue, namely, to consider PES as an adverse event after GVE. The title, the text of the manuscript, and Table 1 have been amended to indicate that PES should be considered as an adverse event. The medical community needs to be aware that a condition such as PES can occur after GVE with coils and understand how this condition should be treated.

As for the pelvic vein thrombosis, the ultimate goal of GVE is to reduce blood flow and to eliminate reflux in the gonadal veins, but not in parametrial or uterine veins. Parametrial and uterine veins are not the target ones in GVE. During the foam sclerotherapy or glue ablation of the gonadal veins (with Glubran 2), the caudal part of the gonadal vein and the pampiniform plexus of the ovary, which is the source of the gonadal vein, are filled with a sclerosant or glue. There is no need to inject the agent into the veins of the uterus and parametria, as with the reduction of blood flow in the gonadal veins, the parametrial and uterine veins shrink in diameter and the reflux in them disappears. This has been proven in our previous study: https://doi.org/10.1016/j.jvsv.2020.05.013

Therefore, there is no need to fill parametrial and uterine veins with sclerosant.

Given that GVE with coils does not directly influence the veins of the parametrium and uterus, we considered the occurrence of thrombi in them as thrombosis of non-target veins, i.e. as a complication. In addition, there is no data on how the thrombotic process develops in parametrial and uterine veins. It is not known whether the thrombus will spread to the tributaries of the internal iliac veins or to the trunk of the internal iliac vein. Due to the fact that thrombosis of these veins is not a predictable event (it occurred in 21% and was absent in 79% of patients), we used anticoagulant therapy to prevent its spread and restore the patency of parametrial and uterine veins as soon as possible.

In addition, thrombosis of the parametrial and uterine veins can also occur after resection of the gonadal veins, as we mentioned in our previous report (https://doi.org/10.1016/j.jvsv.2020.05.013) . Therefore, we also regarded this event as a complication of surgical intervention. An effective alternative is presented here: https://doi.org/10.1016/j.jvsv.2020.05.013 .

The authors would like to note that, of course, the reviewer has his own opinion and his own experience, but the authors also have enough of their own experience and opinion, which are evidenced by this study.

We respect the experience of the reviewer and ask to respect our data as well.

Further research of the issue of pelvic vein thrombosis after GVE will provide us with sufficient information to make a consensus decision, however, it is at least counterproductive to neglect the results obtained by other researchers and their own interpretation.

The authors suggest the reviewer to conduct a collaborative prospective study on this issue and convinced that this will be the best way to resolve our dispute.

I have only made comments on the abstract at present for 2 reasons - firstly, they are not supportable at the current time and are highly biased and secondly, the abstract is very important as this is what most people read - and so must give an accurate impression of the work.

Response: In regard to the abstract, the authors regret that the respected reviewer limited himself to comments only. The text of the manuscript contains information that explains the point of view of the authors. However, the authors have made changes to the abstract and softened some of their positions.

There are several other points in the paper that would need to be addressed if the editors wish to publish the paper.

Abstract:

 Introduction

" The efficacy and safety of gonadal vein embolization (GVE) with coils in the treatment of pelvic venous disease (PeVD) has not been definitively determined, and the outcomes of using this technique do not meet the requirements of doctors and patients" extremely biased comment and not substantiated in the literature. There are a great many papers demonstrating the efficacy of coil embolization, the patient improvement in symptoms and also the safety of the technique, even when performed under local anaesthetic and also in the long term.

Response: Yes, there are many studies in the literature that have demonstrated the efficacy of GVE, but there are also studies in which its efficacy was not so obvious, and the manuscript contain references to these works: https://doi.org/10.1016/j.jvs.2011.01.079; https://doi.org/10.1016/j.ijgo.2013.10.008; https://doi.org/10.1007/s00404-015-3895-7.

Literature data are contradictory, and in the absence of comparative randomized trials, the GVE efficacy still remains to be proven. The rates of complication also vary, therefore, GVE safety has not been proven as well. All these statements are supported in the manuscript by literature data. According to Nasser et al., the complete CPP relief was achieved in only 37% of patients. It's not a very good result, right? In our study, the complete CPP relief was 73 in the short- and 94% in the long-term follow-up period. That is, the short-term GVE efficacy is influenced by PES, but after its relief the efficacy becomes high! No large studies have been performed to compare outcomes after GVE with those after glue ablation or resection of the gonadal veins. At present, there is no ideal treatment for PeVD, and this is obvious. Each technique has advantages and disadvantages.

Results.

The "complications" are worthy of note:

"protrusion of coils (5.3%)" - this is an amazingly high statistic - and something we have never seen in thousands of GVEs - this must be related to the technique used in the author's unit or the device they used and should not be generalized as a widespread complication.

 Response: The technique of GVE was standard, it is described in the Methods section and did not differ in any way from the technique presented in many articles by other authors. The authors did not invent anything in regard to embolization. The incidence of coil protrusion 5.3% is a statistic result. You may agree or disagree with these data, but we will not conceal the obvious complication of embolization, which has been repeatedly described in the literature.

"postembolization syndrome (22%)" is a transient problem that is part of the healing process and does not last into the long term. It is an effect of the inflammation caused by thrombosing veins with coils and is not a complication but a consequence of treatment. Our patients have this included in their consent form and so are expecting it. The discomfort of healing is natural and expected and found in most (if not all) effective procedures - it is not a complication.

 Response: That's true. It is a temporary problem, but why can't this temporary problem be considered as a complication? PES should be treated, as it causes significant discomfort to patients over a long period of time. Coil migration or protrusion is also a temporary problem, and efforts are made to resolve it, like with PES, but for some reason coil migration and protrusion are considered as complications.

The authors proposed a consensus solution to this issue above. The authors have nothing to add on this matter.

"thrombosis of the parametrial, uterine veins" - this is the purpose of GVE. It is also why so many workers in the field use foam sclerotherapy in these vein before deploying coils in the larger veins.

Response: The authors presented their arguments regarding pelvic vein thrombosis above.

The goal of GVE is to reduce blood flow in the gonadal veins, eliminate reflux in the gonadal veins, but not to develop thrombosis of the parametrial and uterine veins. This is the point of view of the authors, and we would like to inform the medical community about this.

If you have an opposite opinion, we treat it with respect.

During foam sclerotherapy, a sclerosing agent fills the pampiniform venous plexus surrounding the ovary, but not the parametrial and uterine veins. We have experience in performing sclerotherapy of the gonadal veins using the “sandwich” technique, and in 2022 we performed 5 embolizations of the gonadal veins using Glubran 2, but in none of the cases did the sclerosant or glue get into the veins of the parametrium and uterus. The results of chemical embolization of the gonadal veins will be presented to the medical community soon.

Reflux in the parametrial and uterine veins disappears 6-12 months after the reduction of blood flow through the gonadal veins https://doi.org/10.1016/j.jvsv.2020.05.013

Conclusion:

"GVE with coils in the treatment of PeVD is associated with a high incidence of complications" - this is not the experience of many of us with many years of experience and high numbers of treatments. If the authors have found this in their own practice, they need to qualify their statement: ie: "In our practice, GVE with coils in the treatment ....."

Response: We have amended the conclusion to make it less strong. This study was carried out by a team of authors, so it is quite natural that it presents the results from OUR practice and not from others.

 "which can be reduced" is an unproven statement unless supported by randomised controlled trials. The authors will need to produce future work to show that their new approaches can indeed improve on coil embolization.

Response: This is an assumption based on the results of the study and the experience of treating patients with different methods. The conclusion has been amended, and this phrase has been removed.

Reviewer 2 Report

A nicely written manuscript.

I have some comments to help improve the study presentation.

This describes a retrospective case series of pelvic vein embolisation - 150 cases in 20 years out of 5000 patients.  This is important as it highlights the relative rarity of the issue.  This represents less than ten patients treated per year.  Interestingly 305 patients were surgically treated and 805 medically so.  I would be extremely interested in these patients especially with the complication profile of embolisation.  Is data available?

The cases with persistent pain and were ascribed to coil perforation (which occurs in most IVC filters without issue) - would these patients not represent potential nickel allergies - easily assessed by whether nickel containing earrings are tolerated.  All current coils contain an amount of nickel.

I would contend that thrombosis of the PV and UV are indeed expected and planned outcomes rather than complications.

how many of the PES patients also had thrombosis of PV and UV?

Are the complication groups exclusive?

i think with these changes a good study will be presented which is important to show the literature of ion issues with this intervention.

Author Response

Reviewer #2

A nicely written manuscript.

I have some comments to help improve the study presentation.

Response: The authors are grateful to the reviewer for a high appraisal of the work and the analysis of our study. Responses to your comments are provided below.

This describes a retrospective case series of pelvic vein embolisation - 150 cases in 20 years out of 5000 patients.  This is important as it highlights the relative rarity of the issue.  This represents less than ten patients treated per year.  Interestingly 305 patients were surgically treated and 805 medically so.  I would be extremely interested in these patients especially with the complication profile of embolisation.  Is data available?

Response: I would like to clarify that there were 1620 patients with PeVD and CPP, of whom 513 were excluded due to the presence of a competing diseases accompanied by CPP. As a result, there were 1107 “clean” patients with PeVD, of whom 305 underwent surgical or endovascular treatment, i.e. pelvic vein interventions were performed in 27.6% of patients (approximately in one third of patients with PeVD). Regarding the rates of interventions on the pelvic veins, you are right, it is no more than 20 patients per year. We use strict criteria for selecting patients for surgical and endovascular treatment. We always prescribe patients with venoactive therapy first and evaluate its efficacy. Then, depending on the prevalence of the pelvic vein abnormalities, we perform one or another method of invasive treatment.

It's not entirely clear what data you mean. Public database or patient registry? There is no such registry. The database was created as part of this study, and we do not have legal rights to make this data publicly available.

The cases with persistent pain and were ascribed to coil perforation (which occurs in most IVC filters without issue) - would these patients not represent potential nickel allergies - easily assessed by whether nickel containing earrings are tolerated.  All current coils contain an amount of nickel.

Response: I would like to clarify that we are talking not about the coil perforation but about coil protrusion, in which the coils protrude excessively through the venous wall, but the integrity of the venous wall is not violated (no perforation).

It was not possible to conduct a retrospective assessment of nickel allergy. Patients' medical records contained indications of intolerance to antibiotics and other medications, but there were no indications of nickel allergy. The lack of a nickel hypersensitivity assessment is noted in the Limitations section. We plan to conduct a series of studies to detect nickel allergy in patients who are scheduled for coil embolization. This will take at least 2 years.

I would contend that thrombosis of the PV and UV are indeed expected and planned outcomes rather than complications.

Response: The authors agree that thrombosis of PVs and UVs is an expected complication. However, this is related to the fact that hemodynamics in the pelvic veins suddenly change. The sudden cessation of venous outflow through the gonadal veins leads to a redistribution of blood flow in the uterus and appendages, slowing down blood flow in the veins of the uterus and parametrium, especially when intervention is bilateral. Pelvic vein thrombosis was found in 21% of patients and was absent in 79%. It is an expected event, but it does not occur in all patients. In our opinion, this is a complication, moreover, a complication that requires the use of special agents - anticoagulants. Anticoagulant treatment is carried out for more than 1 month, and according to the SIR classification, this complication is classified as a severe one (class C).

how many of the PES patients also had thrombosis of PV and UV?

Response: These data are presented in Table 3. Thrombosis of PVs and/or UVs was detected in 7 (21.2%) patients with PES and 25 (21.3%) of patients without PES (P=0.98 between the groups).

Are the complication groups exclusive?

Response: These complications are specific to GVE with coils. Other treatment methods were not evaluated in this study. As Table 3 shows, 7 patients with PES had a concomitant pelvic vein thrombosis. No other combinations of complications were revealed.

i think with these changes a good study will be presented which is important to show the literature of ion issues with this intervention.

Response: Thank you very much for your appraisal.

Reviewer 3 Report

The analysis covered a large group of patients treated over a long period of time, but it is retrospective in nature and this is the greatest limitation of this study. Nevertheless, I agree with the authors that the study results can be used to define indications for the endovascular treatment of PeVD, the differentiated choice of embolization agents and systems for their delivery to the gonadal veins, and the justified rejection of the use of GVE with coils in a patient with PeVD. It should also be remembered that in the last 20 years there has been a huge progress and development in both the diagnosis of PeVD and the treatment methods. It is still an underestimated problem, especially when it comes to chronic pelvic pain, which is why I consider the results of this study very valuable.

Author Response

Reviewer #3

The analysis covered a large group of patients treated over a long period of time, but it is retrospective in nature and this is the greatest limitation of this study. Nevertheless, I agree with the authors that the study results can be used to define indications for the endovascular treatment of PeVD, the differentiated choice of embolization agents and systems for their delivery to the gonadal veins, and the justified rejection of the use of GVE with coils in a patient with PeVD. It should also be remembered that in the last 20 years there has been a huge progress and development in both the diagnosis of PeVD and the treatment methods. It is still an underestimated problem, especially when it comes to chronic pelvic pain, which is why I consider the results of this study very valuable.

Response: The authors are grateful to the reviewer for the appraisal of this study.

Reviewer 4 Report

The subject to the pelvic vein incompetence is an important topic in the current phlebology. The authors descibed the serie of the patients treated by embolisation because of the ovarian vein incompetence. The subject of the study is an analysis of the complications related to the ovarian vein embolisation. Some questions to the authors which should be clarified:

Did the Authors use the sclerotherapy in the combination with coil embolisation? If not, the question is how low into the pelvis the embolisation was preformed ?  - the ovarian veins only or also the periuterine veins ? - did the location of the coils (ovarian veins or pelvic veins) influence on the postembolisation syndrome occurrence? 

The second question concers the periuterine vein thrombosis - is this the complication or should it be the effect of the embolisation of the autflow vessels? In all the cases this was an asymptomatic sequale of the outflow vessel embolisation. It is not clear from the text - did the Authors treat the calf vein thrombosis patients only with anticoagulation or also this group of the subjects with periuterine vein thrombosis ? If Yes, how this treatment influenced in the treatment results. 

The conclusions should not containt % values as presented in the results. I also think that  the sentece "the use of anticoagulant prophylaxis of VTE will help to minimize the number of complications ... for patients with PeV - this was not studied in the reserach. 

Author Response

Reviewer #4

The subject to the pelvic vein incompetence is an important topic in the current phlebology. The authors descibed the serie of the patients treated by embolisation because of the ovarian vein incompetence. The subject of the study is an analysis of the complications related to the ovarian vein embolisation. Some questions to the authors which should be clarified:

Response: The authors are grateful to the reviewer for a high appraisal of the work and the analysis of our study. Responses to your comments are provided below.

Did the Authors use the sclerotherapy in the combination with coil embolisation? If not, the question is how low into the pelvis the embolisation was preformed ?  - the ovarian veins only or also the periuterine veins ? - did the location of the coils (ovarian veins or pelvic veins) influence on the postembolisation syndrome occurrence?

Response: In this study, we did not combine sclerotherapy of the gonadal veins with their coil embolization. A clarification has been added to the Methods section.

The coils were placed caudally as far as possible, i.e. closer to the pampiniform plexus and not higher than the middle third of the gonadal vein. Coils were used to close only the gonadal veins. When sclerotherapy or glue ablation is performed, there is no need to embolize parametrial and/or uterine veins with sclerosants or glue, as these veins shrink in size and reflux disappears in them 6-12 months after coil embolization of the gonadal veins. The main effect of the intervention is hemodynamic. https://doi.org/10.1016/j.jvsv.2020.05.013

The second question concers the periuterine vein thrombosis - is this the complication or should it be the effect of the embolisation of the autflow vessels? In all the cases this was an asymptomatic sequale of the outflow vessel embolisation. It is not clear from the text - did the Authors treat the calf vein thrombosis patients only with anticoagulation or also this group of the subjects with periuterine vein thrombosis ? If Yes, how this treatment influenced in the treatment results.

Response: Thrombosis of the pelvic veins of parametrium and uterus was present in 21% and absent in 79% of patients. Thus, in the post-embolization period the thrombosis of parametrial and/or uterine veins occurred only in every fifth patient. This is not 50% and not 70%. Therefore, thrombosis of parametrial and/or uterine veins is an expected, but absolutely not obligatory event, and, if present, it requires anticoagulant treatment, because we don’t know whether the thrombus will spread further along the pelvic veins to the tributaries and trunk of the internal iliac vein. We cannot control the occurrence of thrombosis of the veins of the parametrium and uterus without using anticoagulants. Due to the fact of thrombosis of non-target veins (gonadal vein embolization is a reduction in blood flow in the gonadal veins, and not the veins of the parametrium and uterus) and the necessity of using the specific medical therapy, we regarded this event as a class C complication, according to the SIR classification.

Patients with deep vein thrombosis of the calf also received anticoagulant therapy, and this event was also regarded as a class C complication, according to SIR. This is indicated in the Results section, in the subsection “Parametrial vein thrombosis…”. An additional clarification has been made in this subsection.

I would like to add that thrombosis of the pampiniform venous plexus, associated with coil embolization, i.e., the venous plexus, which is the source of the gonadal vein, was not considered as a complication. This thrombosis is a required effect of the intervention, enhancing the reduction of venous outflow through the gonadal vein. The authors start thinking if this can be a "terminological trap", especially in translation. We would like to clarify that parametrial veins are veins in the broad ligament of the uterus, uterine veins are veins located in the uterus, and pampiniform venous plexus is a venous plexus that surrounds the ovary and is the source of the ovarian vein. Maybe these terms will clarify the situation better, because we want to convey our data as fully as possible so that there is no misunderstanding related to terminology or translation.

The conclusions should not containt % values as presented in the results. I also think that  the sentece "the use of anticoagulant prophylaxis of VTE will help to minimize the number of complications ... for patients with PeV - this was not studied in the reserach.

Response:

The Conclusion section was amended accordingly.
